# A bifunctional asparaginyl endopeptidase efficiently catalyzes both cleavage and cyclization of cyclic trypsin inhibitors

Junqiao Du[1], Kuok Yap[1], Lai Yue Chan[1], Fabian B. H. Rehm[1], Fong Yang Looi[1], Aaron G. Poth[1], Edward K. Gilding[1], Quentin Kaas[1], Thomas Durek[1✉] & David J. Craik[1✉]

Asparaginyl endopeptidases (AEPs) catalyze the key backbone cyclization step during the biosynthesis of plant-derived cyclic peptides. Here, we report the identification of two AEPs from *Momordica cochinchinensis* and biochemically characterize MCoAEP2 that catalyzes the maturation of trypsin inhibitor cyclotides. Recombinantly produced MCoAEP2 catalyzes the backbone cyclization of a linear cyclotide precursor (MCoTI-II-NAL) with a $k_{cat}/K_m$ of 620 $mM^{-1} s^{-1}$, making it one of the fastest cyclases reported to date. We show that MCoAEP2 can mediate both the N-terminal excision and C-terminal cyclization of cyclotide precursors in vitro. The rate of cyclization/hydrolysis is primarily influenced by varying pH, which could potentially control the succession of AEP-mediated processing events in vivo. Furthermore, MCoAEP2 efficiently catalyzes the backbone cyclization of an engineered MCoTI-II analog with anti-angiogenic activity. MCoAEP2 provides enhanced synthetic access to structures previously inaccessible by direct chemistry approaches and enables the wider application of trypsin inhibitor cyclotides in biotechnology applications.

[1] Institute for Molecular Bioscience, The University of Queensland, Brisbane, Queensland 4072, Australia.
✉email: t.durek@imb.uq.edu.au; d.craik@imb.uq.edu.au

Ribosomally synthesized and post-translationally modified peptides (RiPPs) are an important class of natural products[1]. Their biosynthesis includes various types of post-translational modifications (PTMs), including head-to-tail cyclization[2]. Cyclotides are a group of plant-derived RiPPs that have a special cyclic cystine knot (CCK) structural motif that confers exceptional stability[3]. Together with their natural function as host defense agents[4], cyclotides have shown promise as scaffolds for biotechnological applications. In particular, the *Momordica cochinchinensis* trypsin inhibitor-I/II (MCoTI-I and MCoTI-II) cyclotides are increasingly being exploited as pharmaceutical scaffolds[5–7]. However, the cyclization of synthetic peptides in vitro remains challenging[8]. Therefore, understanding the mechanism(s) behind cyclotide biosynthesis is key for harnessing the potential of cyclic peptides in the agricultural and pharmaceutical industries.

Cyclotides are gene-encoded as linear precursors, where the cyclotide domain is flanked by a N-terminal leader peptide and a C-terminal follower peptide (Fig. 1). This organization necessitates at least two processing events: one at the N-terminus and another at the C-terminus of the cyclotide domain. In plants, asparaginyl endopeptidases (AEPs, also known as vacuolar processing enzymes or legumains) have been shown to recognize a conserved Asn/Asp residue at the C-terminal processing site, and cyclize the cyclotide domain by transpeptidation[9–11]. The AEPs characterized so far from cyclic peptide-producing plants include butelase-1 from *Clitoria ternatea*, OaAEP1$_b$ from *Oldenlandia affinis*, VyPALs from *Viola yedoensis* and HaAEP1 from *Helianthus annuus*[12–15]. However, before cyclization, the leader peptide must be removed, and the enzyme(s) involved in this step remain poorly understood.

Cleavage of the leader peptide by papain-like cysteine proteases was recently shown to be important for the maturation of kalata-like cyclotide precursors[16], but as the N-terminal processing site varies among different precursors (Fig. 1), other proteases might also play a role. As trypsin inhibitor cyclotides have a conserved Asn at their N-terminal site, it is hypothesized that for this subfamily of cyclotides AEPs may be involved in N-terminal processing in addition to the C-terminal cyclization[17]. Indeed, an AEP was recently shown to be involved in the N-terminal processing and cyclization of the sunflower trypsin inhibitor SFTI-1 (refs. [18,19]). Given MCoTI and SFTI-1 precursors both contain an Asn residue at the N-terminal processing site (Fig. 1), we hypothesized the N-terminal of MCoTI precursors might also be processed by an AEP.

Here we report the identification of two AEPs from *M. cochinchinensis*, named *MCoAEP1* and *MCoAEP2*. We show recombinant MCoAEP2 can perform both N-terminal proteolysis and C-terminal cyclization of synthetic MCoTI precursor substrates with high catalytic efficiency ($k_{cat}$ values of 19.86 s$^{-1}$ and $k_{cat}/K_m$ of 620 mM$^{-1}$ s$^{-1}$), making it one of the most efficient cyclases reported to date. We further demonstrate that MCoAEP2 is capable of cyclizing MCoSST-01 (ref. [20]), an engineered anti-angiogenic peptide based on the MCoTI-II scaffold. Our study provides insights into the biosynthesis of *Momordica* cyclotides, and has potential biotechnological applications, particularly for the in vitro production of engineered cyclic trypsin inhibitors.

## Results

**Characterization of AEP activity in vitro.** We first aimed to ascertain whether MCoTI precursors could be processed by an AEP. To characterize AEP-mediated cyclization in vitro, we designed a series of peptide substrates comprising an oxidatively folded cyclotide domain flanked by truncated leader and follower regions (Table 1). The leader and follower sequences in MCoTI precursors each are generally between 15 and 20 amino acids in length, but only two extraneous residues at the C-terminus are required to facilitate the targeting and function of AEPs[11,21]. To simplify chemical synthesis, the designed precursor substrates were kept to fewer than 40 amino acids by shortening the pro-peptide with Ala-Leu as the truncated follower sequence and Asp-Ile-Asn as the truncated leader sequence (based on the conservation of these residues among *TIPTOP* genes)[22]. For comparison, we also prepared a substrate (TIPTOP2 unit 3) with full-length leader and follower sequences (Table 1).

We then tested whether the well-characterized AEP from *O. affinis* (OaAEP1$_b$)[13] could cyclize the synthetic substrate MCoTI-II-DAL (Fig. 2a). In fact, MCoTI-II-DAL was not processed by this AEP, even after extended incubation (20 h) and at high enzyme and substrate concentrations (100 nM enzyme and 100 µM substrate). As OaAEP1$_b$ has been shown to cyclize kB1 precursors with C-terminal NGL motifs[13], we attempted to cyclize kB1-NAL to investigate the influence of the Ala-Leu motif in the follower sequence. OaAEP1$_b$ efficiently cyclized kB1-NAL (Fig. 2b), suggesting that other structural factors besides the follower sequence are responsible for the lack of activity against MCoTI-II-DAL. As OaAEP1$_b$ could not process the C-terminus of the synthetic MCoTI-II precursor, we sought to identify the native AEP cyclase in *M. cochinchinensis* involved in the biosynthesis of cyclic trypsin inhibitors.

**Identification and recombinant expression of MCoAEPs.** We identified two full-length AEP sequences in our transcriptome assembled from a combined dataset consisting of male flower, leaf, root, and seed of *M. cochinchinensis* (BioProject ID: PRJNA531039). The cDNAs coding for AEPs were amplified by

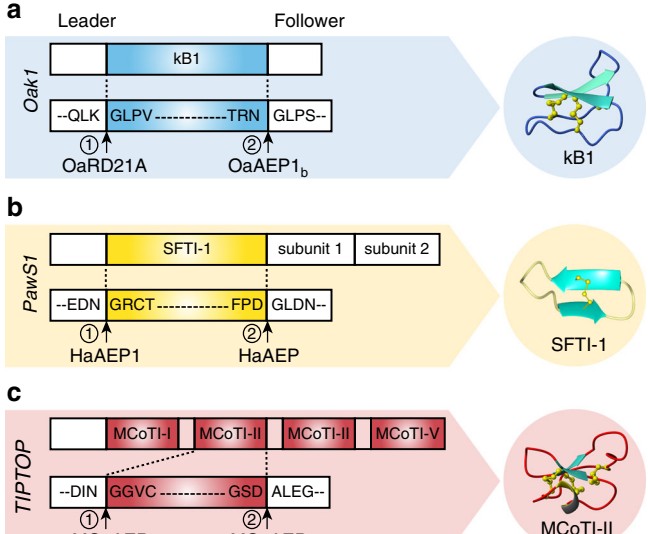

**Fig. 1 Schematic representation of plant cyclic peptide precursors. a** *Oak1* processing is mediated by kalatase A (OaRD21A), which catalyzes N-terminal processing[16], and OaAEP1$_b$ (an AEP from *Oldenlandia affinis*), which targets the C-terminal processing site[13]. **b** The maturation of SFTI-1 from *PawS1* involves HaAEPs (*Helianthus annuus* AEP)[14]. **c** Representative *TIPTOP* precursor encoding several MCoTI-II domains. AEPs might be involved in both the proto-N-terminal cleavage of each MCoTI peptide domain and their cyclization at the C-terminus. Arrows indicate the N- and C-terminal processing sites. Circled numbers indicate the proposed order of processing during biosynthesis. The 3D structures of cyclic peptides are shown next to the precursor from which they originate (PDB IDs for kB1: 1nb1; SFTI-1: 1jbl; MCoTI-II: 1ha9).

**Table 1 Peptide substrates used in this study.**

| Peptide | Sequence | obs. Mass[a] | calc. Mass[a] |
|---|---|---|---|
| MCoTI-II-DAL | GGV-CPKILKKCRRDSDCPGA-CICRGNGYCGSGSD**AL** | 3653.69 | 3653.67 |
| MCoTI-II-NAL | GGV-CPKILKKCRRDSDCPGA-CICRGNGYCGSGSN**AL** | 3652.91 | 3652.70 |
| DIN-MCoTI-II | **DIN**GGV-CPKILKKCRRDSDCPGA-CICRGNGYCGSGSD | 3811.76 | 3811.71 |
| DIN-MCoTI-II-DAL | **DIN**GGV-CPKILKKCRRDSDCPGA-CICRGNGYCGSGSD**AL** | 3995.93 | 3995.83 |
| DID-MCoTI-II-NAL | **DID**GGV-CPKILKKCRRDSDCPGA-CICRGNGYCGSGSN**AL** | 3995.92 | 3995.83 |
| AcDID-MCoTI-II-NAL | **DID**GGV-CPKILKKCRRDSDCPGA-CICRGNGYCGSGSN**AL**[b] | 4037.89 | 4037.83 |
| TIPTOP2 unit 3 | …GGV-CPKILKKCRRDSDCPGA-CICRGNGYCGSGSD…[c] | 6867.20 | 6867.21 |
| MCoSST-01-DAL | GGV-CPKILKKCRRDSDCPGA-CICR*YwKV*CGSGSD**AL** | 3838.90 | 3838.83 |
| kB1-NAL | GLPVCGET—CVGGT-CNTPGCTCSWPV-CTR-N**AL** | 3093.34 | 3093.32 |

Sequences in bold denote the truncated leader peptide and truncated follower peptide. P1 Asx residues are underlined. MCoSST-01-DAL is a grafted peptide based on the scaffold of wild type MCoTI-II, and the italicized residues correspond to the grafted epitope. Lower case w indicates D-Trp
[a]Observed (obs.) and calculated (cal.) monoisotopic mass [M + H]$^+$
[b]N-terminal residue is acetylated
[c]This peptide carries the full-length leader (ALEGLMSDSRAQIDIN) and follower (ALEGLMSDGRAQIDIN) sequences at its N and C termini, respectively

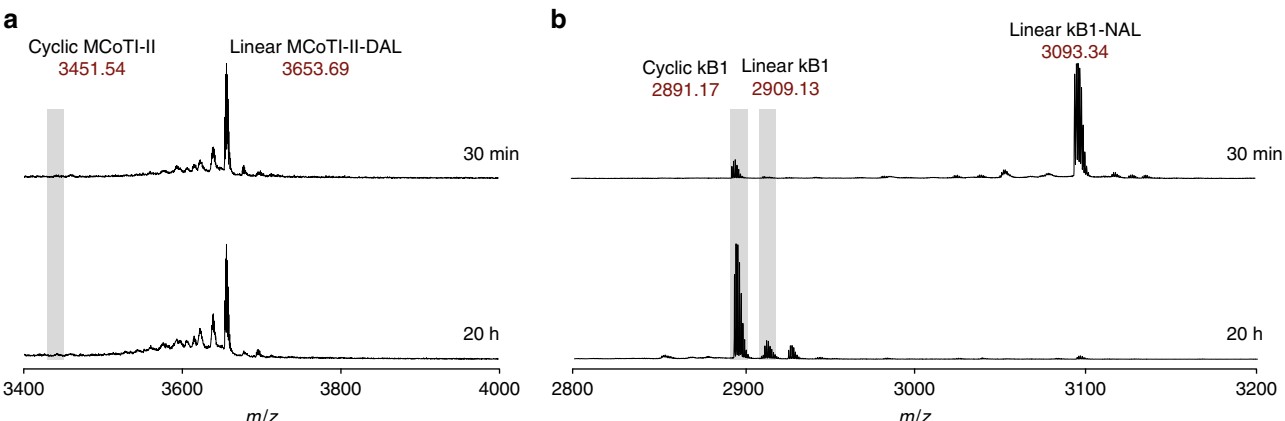

**Fig. 2 Characterization of OaAEP1$_b$ activity against MCoTI-II-DAL and kB1-NAL precursors. a** MALDI-TOF-MS profile of OaAEP1$_b$ (100 nM)-mediated reaction with MCoTI-II-DAL (100 μM). **b** MALDI-TOF-MS profile of OaAEP1$_b$ (100 nM)-mediated reaction with kB1-NAL (100 μM). Both reactions were run in 0.1 M NaOAc, 0.1 M NaCl, 1 mM EDTA, pH 5 at 22 °C, and aliquots were removed after 30 min and 20 h. The $m/z$ region for observed or expected cyclic or linear products is shaded gray.

PCR and cloned for further study (Supplementary Figs. 1 and 2). The sequences were deposited in Genbank with the gene symbols *MCoAEP1* (accession No: MK770254) and *MCoAEP2* (accession No: MK770255). These AEPs share 53.6% sequence identity as determined by pairwise global alignment (www.ebi.ac.uk/Tools/pas/). Compared to other AEPs, MCoAEP1 and MCoAEP2 have a sequence identity of 50.8%, 64.4% with OaAEP1$_b$, 50.5%, 65.6% with butelase 1, and 35.2%, 34.8% with human legumain, respectively. The sequence alignments also suggest the same arrangement of the putative catalytic triad of MCoAEPs (i.e., Asn71, His176, Cys218).

As MCoAEP1 was mainly expressed in insoluble form in *E. coli* (Shuffle® T7 Express), we chose MCoAEP2 over MCoAEP1 for biochemical characterization studies. We successfully expressed MCoAEP2 as a His6-ubiquitin fusion protein (Supplementary Fig. 3). AEPs are typically produced as inactive proenzymes and require auto-activation at low pH (4–6); conditions that are typically encountered in plant vacuoles[23–25]. Therefore, we incubated recombinantly expressed pro-MCoAEP2 (~54 kDa) at pH 4 to produce the mature active enzyme (~32–35 kDa; Supplementary Fig. 3a). The mature enzyme was purified (Supplementary Fig. 3b) and its activity was quantified using the MCoTI-II-DAL synthetic substrate (Supplementary Fig. 3c). We confirmed the activated MCoAEP2 enzyme contains the autocatalytic processing sites at Asp47 and Asn335 by identifying peptide fragments generated from in-gel tryptic digestion

(Supplementary Fig. 3d). Finally, we determined the concentration of active MCoAEP2 through active site titration using the caspase inhibitor Ac-YVAD-CMK (Supplementary Fig. 4).

**MCoAEP2 preferentially cyclizes native MCoTI-II precursors.** Previous reports show AEPs have ligase and/or cyclase activities at a pH of 6–7, whereas hydrolase activity is more prevalent at acidic pH (~4)[14,26]. Therefore, we characterized the effects of pH on the AEP-mediated cleavage and cyclization of the different peptide substrates listed in Table 1. Each substrate (at 50 μM concentration) was tested over a range of pH values, and the initial rate of cyclic product formation determined via quantitative liquid chromatography-mass spectrometry (LC-MS). When incubated with MCoTI-II-DAL, MCoAEP2 functioned as a cyclase with little (if any) hydrolysis product detected by Matrix-Assisted Laser Desorption/Ionization (MALDI)-MS or electrospray ionization (ESI)-QTRAP (Supplementary Fig. 5). The optimal pH for the cyclization reaction was pH 5 (Fig. 3a). Meanwhile, the optimal pH of MCoAEP2 against the non-native peptide precursor kB1-NAL was closer to pH 6 (Fig. 3c). Given this substrate-dependent pH preference of MCoAEP2, pH 5 and pH 6 were selected as optimal conditions in which to determine the Michaelis–Menten parameters of the cyclization reaction catalyzed by MCoAEP2 with MCoTI-II-DAL and kB1-NAL substrates, respectively (Fig. 3b, d). The $k_{cat}$ and $K_m$ values of MCoAEP2 for MCoTI-II-

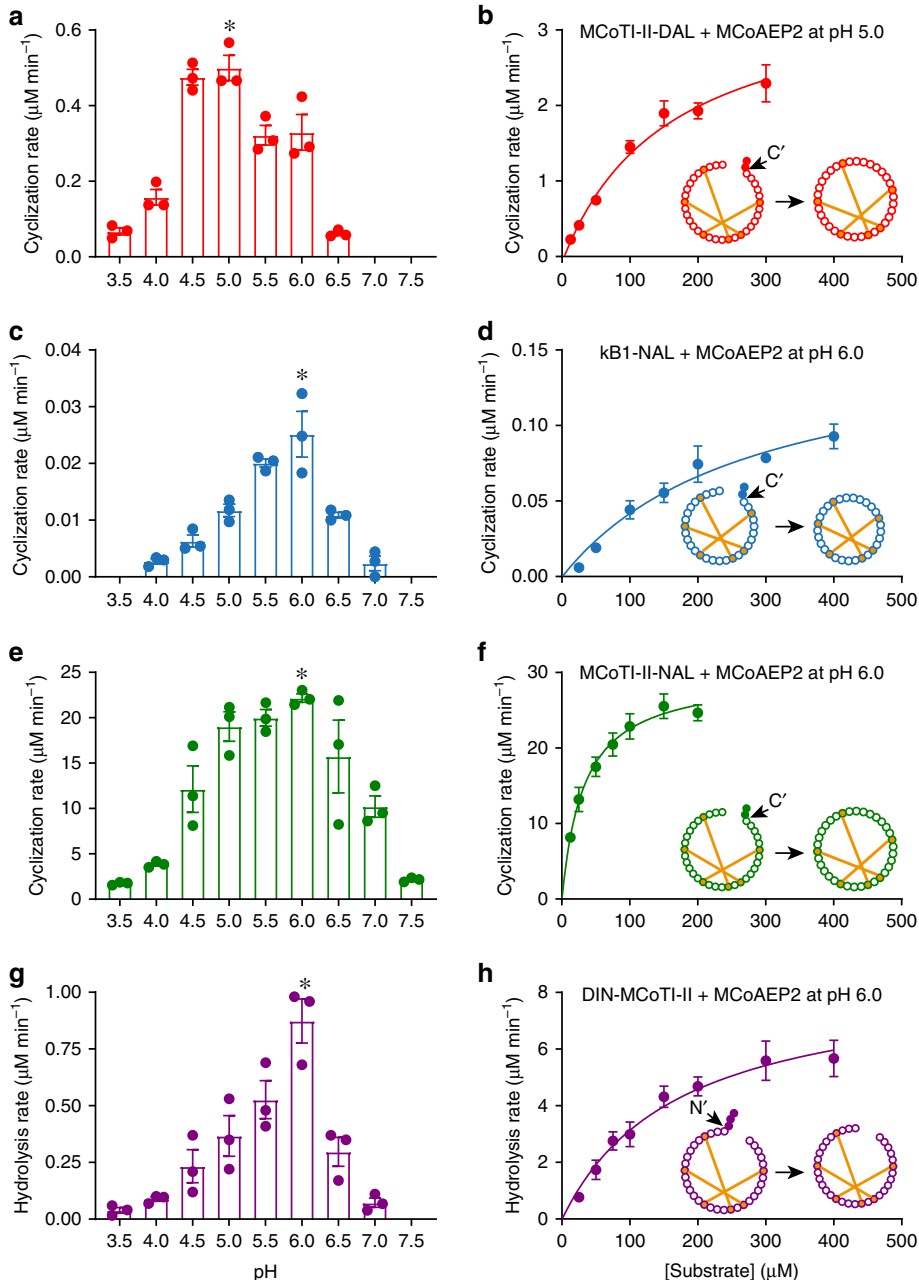

**Fig. 3 pH preference and kinetic characterization of MCoAEP2. a–h** pH preference profile (**a**, **c**, **e**, **g**) and Michaelis–Menten plots (**b**, **d**, **f**, **h**) of MCoAEP2 kinetics for (**a–b**) MCoTI-II-DAL (red), (**c–d**) kB1-NAL (blue), (**e–f**) MCoTI-II-NAL (green), and (**g–h**) DIN-MCoTI-II (purple). The optimal pH is indicated by an asterisk; pH preference experiments were conducted at 22 °C for 5 min in the presence of 25 nM MCoAEP2 with 50 μM of the indicated substrates. Michael–Menten kinetics were performed using 25 nM MCoAEP2 with varying substrate concentrations at optimum pH. Three biological replicates were performed for each experiment (n = 3). All data are presented as means ± SEM.

DAL were $2.40 \pm 0.28\ \text{s}^{-1}$ and $160 \pm 39\ \mu\text{M}$ (mean ± SEM, n = 3), respectively. Meanwhile, MCoAEP2 showed a significantly lower $k_{\text{cat}}$ of $0.02 \pm 0.01\ \text{s}^{-1}$ and a slightly higher $K_{\text{m}}$ of $200 \pm 51\ \mu\text{M}$ (n = 3) for kB1-NAL compared to the native MCoTI-II precursor (Table 2). Therefore, although MCoAEP2 can cyclize both the native MCoTI-II-DAL precursor and the non-native kB1-NAL substrate, it is more efficient at catalyzing precursors comprising a native *Momordica* domain.

The cyclization rate of OaAEP1b against the native MCoTI-II precursor was so low that it was challenging to measure accurately. Therefore, we could only use the kinetic experiments to characterize the relative efficiency of OaAEP1b against the kB1-NAL substrate. We found pH 6.5 was optimal for

OaAEP1b-mediated cyclization of kB1-NAL (Supplementary Fig. 6), with a $k_{\text{cat}}$ and $K_{\text{m}}$ of $0.22 \pm 0.01\ \text{s}^{-1}$ and $34 \pm 6\ \mu\text{M}$ (n = 3), respectively. Together, these findings suggest cyclotide precursors are most efficiently processed by AEPs originating from the same plant.

**Charge interactions may drive enzyme-substrate preference.** We further explored the selective processing of cyclotide precursors by different AEPs at the molecular level via molecular modeling (Fig. 4). We built a homology model of the catalytic domain of MCoAEP2 using the HaAEP1 crystal structure as a template[14]. To identify the potential binding interface, we also

**Table 2 Kinetic parameters of MCoAEP2.**

| Substrate | Act[a] | pH | $k_{cat}$ (s$^{-1}$) | $K_m$ ($\mu$M) | $k_{cat}/K_m$ (mM$^{-1}$ s$^{-1}$) |
|---|---|---|---|---|---|
| MCoTI-II-DAL | C | 5.0 | 2.40 ± 0.28 | 160 ± 39 | 15 ± 7.10 |
| MCoTI-II-NAL | C | 6.0 | 19.86 ± 1.02 | 32 ± 6 | 621 ± 182 |
| DIN-MCoTI-II | H | 6.0 | 5.68 ± 0.67 | 170 ± 41 | 33 ± 16 |
| kB1-NAL | C | 6.0 | 0.02 ± 0.01 | 200 ± 51 | 0.10 ± 0.01 |

[a]Indicates the enzyme's monitored activity: cyclase (C) or hydrolase (H). All data are presented as means ± SEM ($n = 3$).

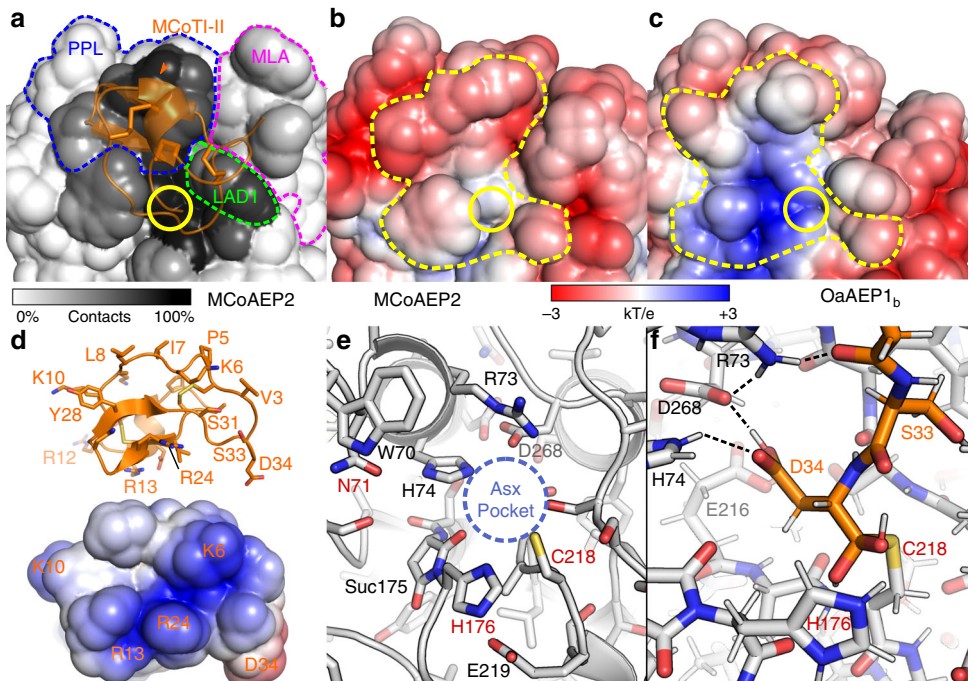

**Fig. 4 Molecular models of MCoAEP2 and OaAEP1$_b$ provide an explanation for selective processing of MCoTI-II-DAL. a** Interaction between MCoTI-II intermediate (C-terminus processed and linked to enzyme) with the surface of MCoAEP2, colored according to the frequency (%) of contacts with MCoTI-II during 1 μs MD simulation. **b** Electrostatic potential mapped on solvent accessible surface of MCoAEP2 apo state molecular model and MCoTI-II interaction area (yellow dashed line delimiting positions with a contact frequency >50% in panel **a**. **c** Electrostatic potential mapped on solvent accessible surface of OaAEP1$_b$ crystal structure (PDB: 5h0i) and MCoTI-II interaction area corresponding to positions identified in panel **a**. **d** Residues of MCoTI-II at the interface and electrostatic potential map of the surface of MCoTI-II at the interface with MCoAEP2 (PDB: 1ha9). **e** Asx pocket and active site of MCoAEP2 in apo state model. **f** Interaction of Asp34 of MCoTI-II in the Asx pocket. Apo MCoAEP2 and the MCoAEP2/MCoTI-II complex were modeled by homology with the HaAEP1 crystal structure (PDB: 6azt) and refined using 1 μs MD simulations. Structures of MCoAEP2 and MCoAEP2/MCoTI-II shown correspond to the centroids of the largest clusters of MD frames determined using a k-means. In panels **a–c**, the Asx binding pocket on the AEP surface is circled in yellow. Regions identified previously as important for the activity of AEPs are highlighted with dashed lines on panel **a**, including the poly-proline loop (PPL) in blue, marker of ligase activity (MLA) in magenta, and ligase activity determinants (LAD1) in green. MCoTI-II is shown in orange in panels **a**, **e** and **f**. The binding of MCoTI-II shown in panel **a** is for illustrative purpose only as several binding orientations were observed during the simulations (Supplementary Fig. 7). The solvent excluded surface in panels **b** to **d** were colored according to the electrostatic potentials computed using APBS 1.4 from −3 kT/e (red) to +3 kT/e (blue). In panel **e**, labels of residues presumably involved in the catalytic reaction are in red; *Suc* stands for succinimide. In panel **f**, hydrogens are shown using narrower sticks and selected hydrogen bonds in the Asx pocket are shown using dashed lines.

modeled a covalently linked MCoTI-II substrate (the C-terminus of which was linked via a thioester bond to the Cys218 side chain of MCoAEP2) in the active site using a 1 μs molecular dynamics simulation (10 simulations of 100 ns starting from different initial binding modes). During the simulations, the MCoTI-II substrate cycled between several binding modes (Supplementary Fig. 7) but, overall, interacted with a limited region of the MCoAEP2 (Fig. 4). Although covalent linkage of the C-terminus of the peptide to the enzyme physically restrained its accessible volume, we found the substrate interacted with several regions previously identified as important for activity in other cyclase AEPs, including: ligase-activity determinant 1 (LAD1)[15], the poly-proline loop (PPL),

and the marker of ligase activity (MLA)[27], but not LAD2 (ref. [15]) (Fig. 4a). The interaction with MLA was only transient, possibly reflecting the large flexibility displayed by this loop during the simulations (Supplementary Fig. 7 and 8). We also found the molecular surface of MCoAEP2 that interacts with the MCoTI-II substrate is mostly electronegative in the apo state (Fig. 4b and Supplementary Fig. 8), in line with the global positive charge of MCoTI-II (Fig. 4d). Meanwhile, the surface of OaAEP1$_b$ is partly neutral or electropositive, indicating a charge incompatibility (Fig. 4c). Overall, our molecular models suggest the differential activity of MCoAEP2 and OaAEP1$_b$ in processing MCoTI-II-DAL arises from charge interactions between the cyclotide

domain and specific regions of the enzymes. Assuming that the catalytic Cys218 is deprotonated in the apo state, which is a requirement for Cys protease catalysis, the electrostatic potential in the proposed binding site of MCoTI-II on MCoAEP2 is predicted to be globally unchanged from pH 3.5 to 7.5 (Supplementary Fig. 9). The electronegative nature of the binding site is therefore suggested to drive the selectivity for MCoTI-II substrate but not pH dependence of processing.

**MCoAEP2 activity is affected by the substrate P1 residue**. The majority of cyclotides (i.e. 1000 out of 1160) display an Asn at the site of cyclization, corresponding to the P1 position (www.cybase. org.au)[28]. In contrast, all cyclic trypsin inhibitors from the *Momordica* family have an Asp at P1 (ref. [29]). To determine the substrate preference of MCoAEP2 at the P1 position (i.e., Asn vs. Asp), we compared its activity against MCoTI-II-DAL and MCoTI-II-NAL, with the latter incorporating an Asn instead of an Asp at P1. MCoAEP2 was significantly more effective at cyclizing MCoTI-II-NAL than MCoTI-II-DAL at all pHs tested (Fig. 3a, e). In addition, the $k_{cat}$ for cyclization of MCoTI-II-NAL (pH 6, Fig. 3f, $k_{cat}$: 19.86 ± 1.02 s$^{-1}$; $n = 3$) with MCoAEP2 is approximately ten times faster than for MCoTI-II-DAL (pH 5, $k_{cat}$: 2.40 ± 0.28 s$^{-1}$; $n = 3$). Although OaAEP1$_b$ was capable of slowly cyclizing MCoTI-II-NAL, the low reaction rates prohibited any quantitative experiments (Supplementary Fig. 10). These results indicate MCoAEP2 preferentially processes cyclic trypsin inhibitors with an Asn at P1.

**Peptidase-type AEPs are weak at recognizing MCoTI-II-DAL**. We also tested the synthetic MCoTI precursors against the well-characterized AEPs from *Helianthus annuus* (HaAEP1), *Arabidopsis thaliana* (AtAEP2) and human legumain[18,30,31], which have all been characterized as predominant peptidase type AEPs. While the native-like MCoTI-II-DAL is a very poor substrate for these enzymes, MCoTI-II-NAL was more efficiently processed yielding correctly cyclized product (Supplementary Fig. 11). This unexpected cyclization capability in these peptidase-type AEPs suggests that the outcome of the enzyme-catalyzed reaction is at least in part substrate-dependent.

**N- and C-terminal processing of MCoTI precursors by MCoAEP2**. To determine whether MCoAEP2 can also catalyze N-terminal processing of MCoTI precursor proteins, we synthesized DIN-MCoTI-II. This synthetic peptide contains a tripeptide leader sequence but lacks any follower peptide residues to avoid interference with cyclizing activity. In the presence of MCoAEP2, acyclic MCoTI-II was produced via cleavage after the Asn residue in the DIN-MCoTI-II substrate as detected by MALDI-MS (Supplementary Fig. 12a). The optimum pH for this reaction was ~6 with a $k_{cat}$ of 5.68 ± 0.67 s$^{-1}$ and a $K_m$ of 170 ± 41 μM ($n = 3$) for the DIN-MCoTI-II precursor (Fig. 3g, h; Table 2). The Michaelis–Menten parameters were comparable to the values obtained for the cyclization of MCoTI-II-DAL ($k_{cat}$: 2.40 ± 0.28 s$^{-1}$, $K_m$: 160 ± 39 μM, $n = 3$), although the pH optimum for the latter reaction was at a more acidic pH of about 5 (Fig. 3a). Therefore, these experiments show MCoAEP2 can efficiently cleave the truncated leader peptide residues of MCoTI precursor proteins in a prelude for cyclization.

To further evaluate enzyme performance, we synthesized the DIN-MCoTI-II-DAL substrate precursor, which carries both a truncated leader peptide and a truncated follower peptide sequence. Upon incubation with MCoAEP2 at pH 5 and pH 6 for 30 min, MALDI-MS indicated production of cyclic MCoTI-II, along with the MCoTI-II-DAL intermediate, and a misprocessed product, cyclic DIN-MCoTI-II (Supplementary Fig. 12b). We also

analyzed the processing of a recombinantly produced TIPTOP2 MCoTI-II precursor (TIPTOP2 unit 3) carrying the full-length leader and follower sequences by MCoAEP2 and compared it to DIN-MCoTI-II-DAL under identical conditions. Both precursors were processed with similar kinetics, indicating that the additional leader and follower residues in the native precursors do not significantly affect the cyclization kinetics (Supplementary Fig. 13).

**Order of N- and C-terminal processing events**. The conservation of recognition residues (Asn/Asp) at both N- and C-terminal processing sites of *TIPTOP* precursors (Supplementary Fig. 14) suggests selective pressure for a preserved processing mechanism. To further investigate the order of processing at N- and C-terminal sites we swapped the P1 Asn/Asp residues and created DID-MCoTI-II-NAL as well as an N-terminally acetylated variant thereof (AcDID-MCoTI-II-NAL). When incubated with MCoAEP2, AcDID-MCoTI-II-NAL is efficiently hydrolyzed at the C-terminus resulting in linear AcDID-MCoTI-II-N (Supplementary Fig. 12c). Under the same conditions, DID-MCoTI-II-NAL produced multiple products but did not yield correct cyclic MCoTI-II that is obtained from the DIN-MCoTI-II-DAL substrate (Supplementary Fig. 12b, d). These results indicate that the placement/conservation of Asn/Asp at the N- and C-terminal processing sites in *TIPTOP* precursors is not random, but likely governs the order of processing at these sites.

**MCoAEP2 can cyclize an engineered MCoTI-II scaffold**. AEPs (like butelase 1 and OaAEP1$_b$) have previously been shown to have a wide range of biotechnological applications, including peptide/protein ligation, peptide macrocyclization, and cell surface labeling[32–37]; however, their utility in grafted peptide production is unclear[38]. As MCoAEP2 can produce native MCoTI-II in vitro, we hypothesized this enzyme could function as an ideal cyclase for grafted peptide production. An example of grafted cyclic peptide from the literature (MCoSST-01, an anti-angiogenic grafted cyclic peptide)[20] based on the MCoTI-II scaffold was chosen as the substrate for MCoAEP2 to test this hypothesis.

A linear MCoSST-01 precursor protein bearing the therapeutic epitope with an Ala-Leu dipeptide at the C-terminus was synthesized (MCoSST-01-DAL, see Table 1). Based on the abovementioned results, enzymatic cyclization of MCoSST-01 was conducted overnight (20 h) at pH 5. We found the enzymatically (MCoAEP2)-cyclized MCoSST-01 product had the same retention time as that of the chemically-synthesized version on reversed phase high-performance liquid chromatography (RP-HPLC; Fig. 5a). In addition, two-dimensional NMR spectroscopy confirmed that the enzymatically-cyclized MCoSST-01 is structurally identical to the chemically-synthesized peptide (Fig. 5b). Therefore, our data indicate that MCoAEP2 can be used to effectively cyclize engineered MCoTI-II scaffolds.

**Discussion**
This study describes the cloning of two AEPs from *M. cochinchinensis*, *MCoAEP1* and *MCoAEP2*, and the recombinant expression and functional and kinetic analyses of the latter. Following recombinant expression and purification, complete activation of MCoAEP2 was achieved by incubation at pH 4. This is consistent with other reports including studies of human and plant legumains that require self-processing at acidic pH to mature into a fully active form[14,24,39]. We also found the auto-processing sites (Asp47 and Asn335) and arrangement of the putative catalytic triad (Asn71, His176, Cys218) of MCoAEP2 are similar to other previously characterized AEPs from *O.*

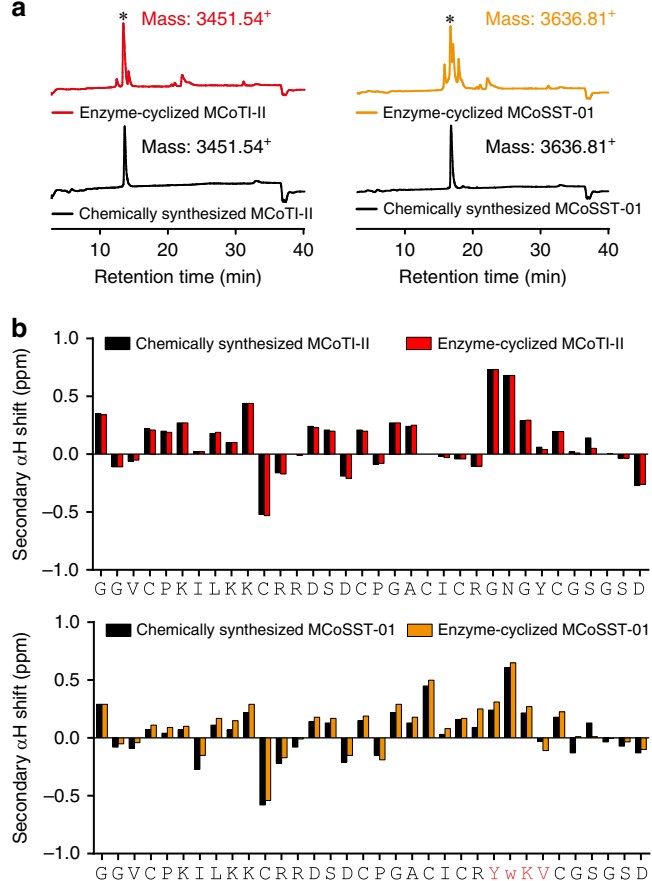

**Fig. 5 Comparison of cyclic products produced via chemical synthesis or MCoAEP2-mediated enzymatic cyclization. a** RP-HPLC profiles illustrate equivalent retention times of enzyme-cyclized products and authentic chemically-synthesized peptides. The peaks corresponding to the target cyclic products are indicated by asterisks. **b** Comparison of NMR secondary Hα chemical shifts of chemically-synthesized vs enzyme-cyclized peptides (red for MCoTI-II, orange for MCoSST-01). MCoSST-01 is a grafted peptide based on MCoTI-II as a scaffold, where the residues in red identify the grafted epitope. Enzyme assays were performed over 20 h at 22 °C, pH 5 in the presence of 50 nM MCoAEP2 and 100 µM substrate.

affinis[13,40], *C. ternatea*[12], jack bean[41], sunflower[14], *Viola* species[36], and human[23,24]. Furthermore, we showed recombinantly produced MCoAEP2 not only efficiently cyclized a linear MCoTI-II-DAL precursor but also a grafted cyclic peptide based on the MCoTI-II scaffold. The ability of MCoAEP2 to cyclize the grafted MCoTI-II suggests the potential application of this enzyme for other reported grafted MCoTI peptides[42] and enables approaches for heterologous production of reengineered MCoTI peptides in planta[43].

Attempts to classify AEPs according to their preferential hydrolase/ligase activities have resulted in several regions located near the active site being proposed as markers of either activity (Fig. 6). For example, Tam and colleagues proposed LAD1 and LAD2 as well as the gatekeeper residue[44] (located within LAD1) as crucial for AEP ligase activity[15]. Jackson et al. proposed a different region: the MLA, which comprises five amino acids whose deletion correlates with preferential ligase activity[27]. Interestingly, MCoAEP2 is also predicted to be a hydrolase based on sequence similarities in the MLA, LAD1, and LAD2 regions alone (Fig. 6). Furthermore, the MCoAEP2 gatekeeper residue is a glycine, which is strongly indicative of hydrolase activity according to functional data reported to date[15,44]. Therefore,

other factors besides the abovementioned regions must contribute to the preferential activity of AEPs. For example, Tam and co-workers recently demonstrated that the pH can alter the activity of a particular AEP such as VyAEP1 from acting as a ligase (more neutral pH) to a hydrolase (more acid pH)[15]. Our observation that AEPs previously classified as peptidases (hydrolases) can efficiently cyclize MCoTI precursors (Supplementary Fig. 11) strongly suggests that the nature of the substrate also contributes to the outcome of the AEP-mediated reactions.

Although we found no significant change in the ratio of hydrolysis vs ligation among our test substrates, our data indicate AEPs can be quite stringent in terms of substrate specificity. We found OaAEP1$_b$ was unable to process MCoTI-II cyclotide precursors but showed good activity against kalata B1 precursors, and the reciprocal activity was observed using MCoAEP2. Therefore, OaAEP1$_b$ and MCoAEP2 appear to preferentially cyclize their native substrates. With this finding, we hypothesize that AEPs may have co-evolved with their substrates. Testing of this hypothesis in future work may provide data that could underpin strategies for identifying AEPs with, for example, altered substrate preferences for biotechnological applications. The activities of these enzymes against cyclotide precursors from other families remain to be determined. Our homology modeling data suggest that the interaction of MCoTI-II intermediate covalently linked to MCoAEP2 could provide an understanding of the molecule specificity which is responsible for the AEP activity. The different surface charges between MCoAEP2 and OaAEP1$_b$ could help explain the preference of MCoAEP2 for MCoTI substrate (positive charge) rather than kB1 substrate (neutral).

We found the catalytic rates of MCoAEP2 spanned three orders of magnitude (0.02–19.86 s$^{-1}$) across the various CCK-constrained peptide substrates trialed in this study, suggesting the kinetics are substrate-dependent. The catalytic efficiency of MCoAEP2 against MCoTI-II-NAL ($k_{cat}/K_m$: 621 mM$^{-1}$ s$^{-1}$) was also significantly higher than other reported cyclases, such as OaAEP1$_b$ with kB1-NAL ($k_{cat}/K_m$: 7.1 mM$^{-1}$ s$^{-1}$; current study), butelase 1 with kB1-NHV ($k_{cat}/K_m$: 10.7 mM$^{-1}$ s$^{-1}$)[12], and comparable to values obtained for butelase 1 against a non-CCK neuromedin U substrate ($k_{cat}/K_m$: 1300 mM$^{-1}$ s$^{-1}$)[45]. Together, our findings indicate MCoAEP2 is one of the fastest cyclases reported to date for processing CCK-constrained substrates.

AEPs prefer substrates displaying an Asn over Asp residue at the P1 position[46,47], and the nature of the P1 residue correlates with different pH optimum. MCoAEP2 displayed the highest efficiency in cyclizing MCoTI-II-DAL (Asp at the P1 position) at pH 5, whereas pH 6 was optimal for cyclization of kB1-NAL and MCoTI-II-NAL precursor substrates (with Asn at the P1 position). This suggests that AEP preferentially recognizes a neutral (uncharged) P1 side chain, and the more acidic pH optimum of Asp vs Asn likely reflects the protonation state of Asp side chains (i.e., pKa of ~4.0). Although MCoAEP2 displays a clear activity preference for Asn in the P1 position, it is much more efficient at cyclizing MCoTI-II-DAL compared to kB1-NAL, which further confirms that the substrate itself is more crucial than P1 position in terms of the catalytic activity.

Despite the observation of MCoAEP2 as a cyclase, this enzyme has also played a key role for leader peptide release where it is capable of generating the linear product when tested against the MCoTI precursor with a truncated leader peptide. Interestingly, the enzyme did not recognize other potential sites within this substrate (e.g., G**N**G**Y**), probably due to the constrained structure preventing close access to the enzyme. Notably, the enzyme plays its bifunctional role in the same in vitro condition resulting in an expected cyclic product when incubated with DIN-MCoTI-II-DAL. This bifunctional role of MCoAEP2 bears some similarity

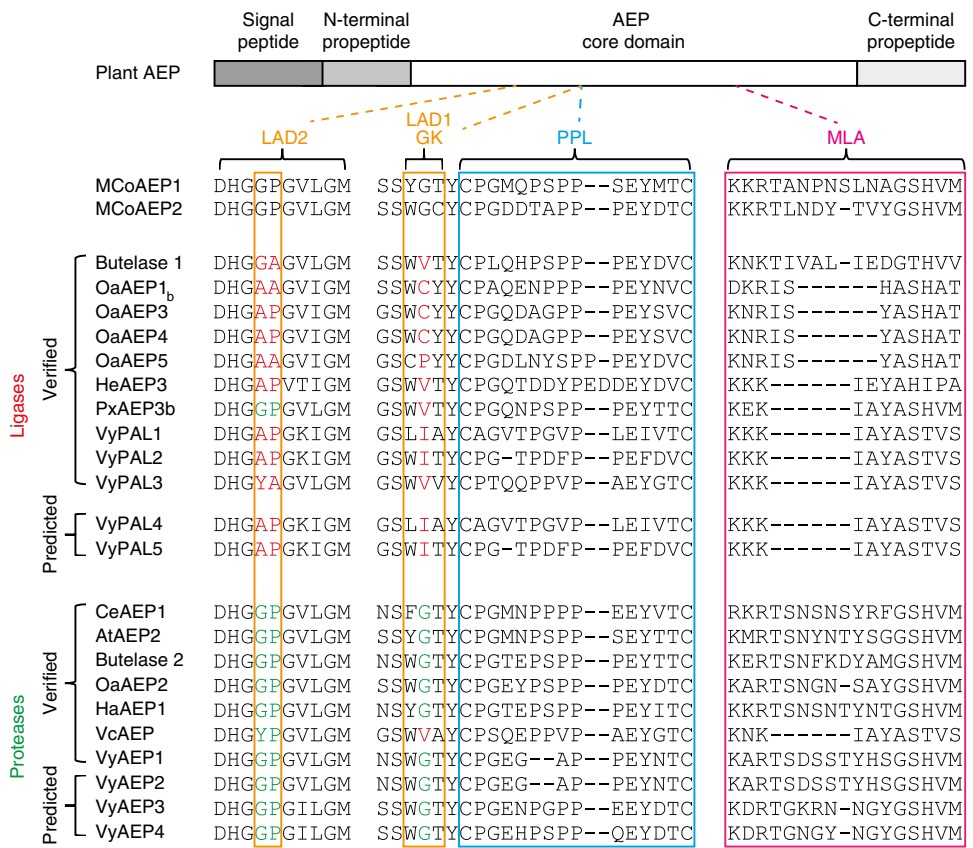

**Fig. 6 Sequence alignment of AEPs. Proposed ligase-activity determinant (LAD) residues are boxed in orange.** Residues from LAD2 or LAD1 gatekeeper (GK) are in red (ligase favored) and green (protease favored), respectively. The poly-proline loop (PPL) is demarcated by a blue box and the region defined as a marker of ligase activity (MLA) is boxed in magenta. AEP activity was either predicted based on sequence similarity or experimentally verified in vitro and/or in planta.

to that described for GmPOPB from the autumn skullcap mushroom (*Galerina marginata*), which can perform N-terminal cleavage and subsequent cyclization steps to produce cyclic amatoxins[48,49].

Interestingly, we observed a mis-cyclized DIN-MCoTI-II that resulted from premature cyclization of the precursor prior to release of the leader peptide. This misprocessed cyclic peptide might be an artifact due to the artificially short leader peptide sequences used in this synthetic peptide precursor, but the absence of misprocessed MCoTI-II variants in planta[22] suggests that plants rapidly degrade these misprocessed variants, or have strategies to ensure timely processing of cyclotide precursors. The latter could be realized through the action of additional AEPs or other proteases in planta that possibly cooperate with MCoAEP2 for efficient processing of the leader peptide of MCoTI precursors to achieve the final cyclic products. We note that the leader peptide of MCoTI precursors (DIN) is highly conserved across various MCoTI natural homologs (Supplementary Fig. 14), suggesting a selective pressure for a preserved processing mechanism at the N-terminus. Additionally, the leader peptide processing site of all natural MCoTI precursors is always an Asn; this N-terminal Asn is processed faster by AEPs than Asp, which is always present at the C-terminal cyclization site (Supplementary Fig. 14). When these conserved P1 residues are swapped, MCoAEP2 preferentially processed the C-terminal Asn site, resulting in a hydrolyzed linear product that could not be processed further or a small fraction of mis-cyclized precursor (Supplementary Fig. 12). This finding suggests that the order of processing is important for native MCoTI-II cyclization and probably controlled by the conserved P1 residue (Asn or Asp). We found MCoAEP2

processes the Asn site substantially faster and more efficiently than an Asp at a more neutral pH (Fig. 3), indicating that the order of processing may also be pH controlled. Therefore, the succession of AEP-mediated processing events could be controlled in vivo through compartmentalization, with the precursor transiting into an increasingly more acidic secretory pathway. Taken together, our results suggest MCoAEP2 is capable of catalyzing both N- and C-terminal maturation steps during cyclotide biosynthesis, but additional factors are likely to control the precise order of processing.

In conclusion, MCoAEP2 is a member of the legumain family capable of both cleaving and cyclizing cyclic precursor proteins. It is one of the fastest cyclases described to date, and efficiently cyclized a reengineered peptide based on the MCoTI-II scaffold. Therefore, MCoAEP2 is a promising cyclase alternative to support cyclotide grafting in trypsin inhibitors, with potential applications in the agricultural and pharmaceutical industries.

## Methods

**Peptide substrates and inhibitors**. The peptide substrates including the fluorogenic peptide substrate Abz-STRNGLPS-Y(3NO₂) (o-aminobenzoic acid [Abz]-Ser-Thr-Arg-Asn-Gly-Leu-Pro-Ser-3-Nitro-L-tyrosine) and other enzymatic substrates were synthesized chemically by standard F$_{moc}$ solid-phase peptide synthesis. The TIPTOP2 unit 3 substrate was recombinantly expressed in *E. coli* as described below. The caspase inhibitor Ac-YVAD-CMK was purchased from Sigma-Aldrich. Peptides were cleaved by a mixture of 95% trifluoroacetic acid (TFA)/2.5% triisopropylsilane/2.5% H₂O. TFA was removed by rotary evaporator and chilled diethyl ether used to precipitate the crude peptide. Peptides were then oxidized and folded in 0.1 M ammonium bicarbonate (pH 8.5) overnight and purified by RP-HPLC with a series of Phenomenex C18 columns to high purity (>95%).

**Production of recombinant TIPTOP2 unit 3.** A synthetic codon-optimized DNA encoding *TIPTOP* unit 3 was cloned into the pHUE-ubiquitin expression vector[50]. This vector contains a His6 tag for affinity purification, a ubiquitin domain as a fusion protein, a tobacco etch virus (TEV) protease recognition site followed by the *TIPTOP* unit 3 gene. The plasmid was transformed into *E. coli* strain BL21(DE3). Cultures were grown at 37 °C in Luria-Bertani medium to mid-log phase (OD: 600 nm ~0.8) and induced with 0.4 mM IPTG for overnight growth at 18 °C. Cells were harvested by centrifugation (6000*g*) and resuspended in lysis buffer (20 mM Tris-HCl, 300 mM NaCl, 10% glycerol, 0.5% Chaps, 10 mM imidazole, pH 8.0). The resuspension was then lysed by a cell disruptor at 32k psi. Cell debris was removed by centrifugation (40,000*g*) and the supernatant was purified by passing over a Ni-NTA (Qiagen) resin. The fusion protein was then eluted with 300 mM imidazole, followed with buffer exchange to remove imidazole. Approximately 200 μg of TEV protease and 2 mM DTT were added and the cleavage reaction was left for overnight at 4 °C. The liberated recombinant peptide was then purified via RP-HPLC and oxidatively folded and purified as described above.

**RNA-seq and transcriptome analysis.** Root, leaf, flower, and seed, of *M. cochinchinensis* were chosen for RNA-seq on an Illumina HiSeq2500 at the Australian Genome Research Facility (Melbourne, Australia) in a paired-end format. RNA was extracted using TRIzol reagent (Life Technologies) according to the manufacturer's standard protocol. Data were deposited in the NCBI-SRA service under project accession PRJNA531039. Data were trimmed using Trimmomatic v0.36 to remove leftover 5' adaptor sequence and bases with a Phred score below Q30 (ref. [51]). Trimmed data were then assembled with Trinity v2.4.0 using default parameters[52]. Finally, AEPs were discovered in the transcriptome assembly by tblastn using OaAEP1ₐ (GenBank ID: ALG36103.1) protein sequence as a query[53].

**MCoAEP cloning.** The flowers of *M. cochinchinensis* were freshly collected from the University of Queensland glasshouse. Total RNA was isolated with TRIzol (Life Technologies) and reversed transcribed with SuperScript III reverse transcriptase (Life Technologies) according to the manufacturer's instructions. Primers were designed with Gateway™ attB sites based on transcriptome data previously acquired in house as follows: *MCoAEP1* (Forward: 5'- GGG GAC AAG TTT GTA CAA AAA AGC AGG CTA TGA CTC GTA TCC CCA ACG GAG -3', Reverse: 5'-GGG GAC CAC TTT GTA CAA GAA AGC TGG GTT CAA GCA GTG AAG CCC TTG TGC -3') and *MCoAEP2* (Forward: 5'-GGG GAC AAG TTT GTA CAA AAA AGC AGG CTA TGG CCG CCC TGA CTC TG-3', Reverse: 5'-GGG GAC CAC TTT GTA CAA GAA AGC TGG GTT CAA GCA CTA AAA CCA CCT TCA TGC-3'). Each primer set generated one AEP sequence and then PCR products were purified and cloned into the pDS221 vector. The pDS221 vector was constructed from pDNOR221 (Invitrogen) except the *nptII* cassette conferring kanamycin resistance in *E. coli* was replaced (using the NEBuilder HiFi kit; New England Biolabs) with a cassette encoding *aadA* from pCR8-TOPO (Invitrogen) to give a vector with spectinomycin resistance. Sequences were cloned into pDS221 by BP gateway cloning using BP Clonease II (Invitrogen). Purified plasmids from colonies were sent for Sanger sequencing at the Australian Genome Research Facility (AGRF; www.agrf.org).

**Recombinant expression of AEPs.** The coding regions of *MCoAEP1*, *MCoAEP2*, *OaAEP1ₐ* (*Oldenlandia affinis*), *HaAEP1* (*Helianthus annuus*), *AtAEP2* (*Arabidopsis thaliana*) and human *legumain* without signal peptides were inserted into the same vector used for recombinant *TIPTOP2* unit 3 but without the TEV recognition sequence. The constructs were confirmed by Sanger sequencing. Plasmids were transformed into the *E. coli* Shuffle® T7 expressing cell strain (New England BioLabs), and grown at 30 °C in LB broth to mid-log phase (OD: 600 nm of 0.6~0.8); the temperature was then decreased to 16 °C and isopropyl β-D-thiogalactoside (IPTG) was added for overnight induction at a final concentration of 0.4 mM. Cells were harvested by centrifugation (6000*g*) and resuspended in the same lysis buffer. The resuspension was then lysed by a cell disruptor at 32k psi. Cell debris was removed by centrifugation (40,000*g*) and the supernatant was analyzed immediately.

**Purification and activation of AEPs.** The lysate containing each expressed AEP was mixed with 3 mL of Talon® metal affinity resin (Clontech) at 4 °C for 1 h, then loaded onto a gravity column. The column was washed with lysis buffer containing 20 mM imidazole, and the target protein was eluted with buffer containing 150 mM and 300 mM imidazole. The fractions were screened by SDS-PAGE and stained with InstantBlue (Expedeon).

To self-activate AEPs, fractions were desalted by a PD10 column (GE healthcare). Then, activation was performed at pH 4.0 with 0.1 mM Tris (2-carboxyethyl) phosphine hydrochloride (TCEP) at 37 °C for 30 min (MCoAEP2) or 2 h (all other AEPs).

MCoAEP2 was subsequently purified by cation exchange chromatography (GE Healthcare) with a linear gradient (0–100% buffer B, 50 mM sodium acetate, 1 M NaCl, 10% glycerol, pH 4.0), and the final product analyzed by SDS-PAGE followed by in-gel digestion for MS/MS confirmation. Purified MCoAEP2 was stored at -80 °C until further analysis.

**Active site titration of MCoAEP2.** Active site titration was carried out to accurately quantify active MCoAEP2. The active MCoAEP2 was diluted 200-fold in reaction buffer (50 mM NaOAc, 50 mM NaCl, 1 mM EDTA, pH 5) and incubated with different concentrations of Ac-YVAD-CMK inhibitor (two-fold serial dilutions, ranging from 10 nm to 0.15625 nM), before the fluorescent peptide substrate Abz-STRNGLPS-Y(3NO₂) was added at a final concentration of 15 μM. Fluorescence intensity was monitored for 25 min (reading interval: 30 s) by a Tecan M1000 microplate reader using excitation/emission wavelengths of 320/420 nm. Assays were performed in triplicate. The inhibited versus uninhibited initial rates were plotted for each inhibitor concentration, and the X-intercept determined by extrapolating the linear portion of the curve, which represents the active site concentration.

**pH preference and enzyme kinetics assay using QTRAP-MS.** To determine the pH preference, 25 nM of MCoAEP2/OaAEP1ₐ was incubated with 50 μM of substrates at 22 °C for 5 min. After each minute, 2 μL of the reaction solution was collected and quenched in 18 μL buffer (40% acetonitrile, 0.5% trifluoroacetic acid, 0.1% formic acid). Then, 2 μL of the quenched reaction was loaded onto a Phenomenex Kinetex C18 column heated to 60 °C by a Sciex Exion UPLC and coupled to a SCIEX 6500⁺ QTRAP-MS (triple quadrupole) to quantify the final product via multiple reaction monitoring analysis (Analyst v1.6.3, Multiquant 3.02).

Enzyme kinetic experiments were conducted at the optimal pH determined for each enzyme-substrate combination. Each substrate was assayed at a range of concentrations between 25 and 400 μM. The same time points and sample volumes were quenched and analyzed as above. $K_m$ and $V_{max}$ were analyzed with the Michaelis–Menten equation using GraphPad Prism 7. $k_{cat}$ was calculated according to the $V_{max}$ and known enzyme concentration. All other cyclization assays were conducted with OaAEP1ₐ (0.1 mM/1 μM) and MCoTI-II-DAL (100 μM)/MCoTI-II-NAL (50 μM) and analyzed by MALDI-TOF/MS (ABI 4700) at different time-points (i.e., 30 min, 1 h, and/or 20 h).

**Activity assays.** To determine the activities of AEPs on MCoTI-II-DAL and MCoTI-II-NAL substrates, the recombinantly expressed AEPs were incubated with 50 μM of each substrate at 22 °C for 30 min, 1 h, and 20 h at pH 5 and pH 6, respectively. To analyze N-and C-terminal processing, linear substrates (50 μM), including DIN-MCoTI-II, DIN-MCoTI-II-DAL, DID-MCoTI-II-NAL, and AcDID-MCoTI-II-NAL were incubated with MCoAEP2 (50 nM) in assay buffer (0.1 M NaOAc, 0.1 M NaCl, 1 mM EDTA) at pH 5 or pH 6, respectively. The enzyme reaction was performed for 30 min at 22 °C and desalted before MALDI-TOF analysis.

**Depletion rates for TIPTOP2 unit 3 and DIN-MCoTI-II-DAL.** To test for possible effects of additional residues at both termini on the activity of MCoAEP2, assays against substrates with both truncated and full-length termini were performed at 50 μM concentration of substrates with 50 nM of enzyme incubated at 22 °C for multiple time points (0, 0.5, 1, 2, 4, 6 h) at pH 5. The depletion rates were monitored via triple quadrupole MS and the depletion curve was fit with one-phase decay parameters.

**Cyclization of the MCoSST-01 grafted peptide.** A grafted peptide based on the MCoTI-II scaffold was chosen and synthesized as a linear version with additional Ala-Leu residues at the C-terminus. The cyclization assay was undertaken at 22 °C for 20 h. The final product produced by MCoAEP2 was screened by RP-HPLC with a linear gradient of buffer B (10 to 40% acetonitrile, 0.05% formic acid) for retention time comparison with the chemically cyclized version. The linear MCoTI-II-DAL substrate was analyzed in parallel.

**NMR spectroscopy.** The correct folding of all substrates used in this study was verified by NMR (see Supplementary Fig. 15). For cyclic peptides analysis, two-dimensional total correlated spectroscopy (TOCSY) and nuclear Overhauser effect spectroscopy (NOESY) spectra were acquired on a Bruker Avance 600 MHz spectrometer at 298 K. All data assignment were done by CCPNMR Analysis 2.4.1 (ref. [54]).

**Molecular modeling.** The three-dimensional structure of the catalytic domain of MCoAEP2 was modeled by homology using Modeller 9v21 (ref. [55]) with the HaAEP1 crystal structure as a template (PDB: 6azt)[14]. The MCoAEP2 sequence aligns without gaps to that of the HaAEP1 structure, apart from the MLA region, which is not resolved in the HaAEP1 structure. A hundred models were computed, and the model with the lowest DOPE score[56] was minimized using 10,000 steps of steepest descent and then conjugate gradient algorithms using pmemd from the Amber 18 package[57] with the ff14SB force field[58].

The crystal structure of HaAEP1 displays a post-translational modification of Asp177 into a succinimide residue, and this modification was modeled in MCoAEP2. The succinimide and following His residue were modeled as a single residue, which was parametrized in the GAFF force field[59] using the antechamber script from the Amber 18 package with default parameters.

The protonation state of side chains at pH 5.5 was predicted using Propka 3.1 (predictions were averaged over the ten models with best DOPE scores)[60] apart from the side chain of Cys218 (for apo state). Propka has been shown to provide accurate pKa predictions in most cases[61,62]; however, it did not correctly predict the protonation state of Cys218 (a deprotonated Cys281 is required for the catalytic mechanism of Cys proteases). Indeed, implicit solvation methods such as Propka were recently shown to be inaccurate at predicting the pKa of cysteine residues[63]. Therefore, we modeled the side chain of Cys218 as unprotonated.

The orientation of the MCoTI-II precursor covalently linked to MCoAEP2 (MCoTI-II-MCoAEP2) was studied by molecular dynamics simulation. A hundred molecular models of MCoTI-II-MCoAEP2 were generated using Modeller using the crystal structure of HaAEP1 and an NMR solution structure of MCoTI-II[64] (PDB: 1ha9) as the templates. HaAEP1 was crystalized with a covalently linked substrate peptide ANN. For the homology modeling, the C-terminus of linear MCoTI-II was aligned with the ANN peptide substrate in the HaAEP1 structure, whereas the structure of termini were not modeled using the MCoTI-II solution structure. The ten models with the lowest DOPE score were then each studied by 100 ns molecular dynamics simulations using Amber 18. The force field parameters for the linker between the enzyme and the peptide were determined as above for the succinimide using antechamber. Water molecules from the HaAEP1 crystal structure were transferred to the initial molecular models of MCoTI-II-MCoAEP2 (apart from the ones causing steric clashes). The systems were then solvated in a truncated octahedral box with at least 1 nm between the solute atoms and the box sides. Sodium and chloride ions were added to reach neutrality and 150 mM NaCl concentration. The systems were then relaxed using 10,000 steps steepest descent minimization and equilibrated over 5 ns by first constraining the position of the protein solute atoms to their initial position, while progressively raising the temperature to 300 K and equilibrating the volume to reach a pressure of 1 atmosphere. The restraints were then progressively released, on the side chain atoms first and then on the main chain atoms over a further 5 ns. Each system was then studied over 100 ns unrestrained molecular dynamics simulations using the Monte Carlo barostat and the Langevin thermostat, to maintain pressure to 1 atmosphere and temperature to 300 K, respectively. The particle-mesh Ewald method was used to model long-range electrostatic interactions. The frames generated for simulations of the MCoAEP2 apo state or of the complex between MCoAEP2 and MCoTI-II were clustered using a k-means as implemented in AmberTools19 cpptraj.

Electrostatic potential generated by MCoAEP2 was computed using APBS 1.4 (ref. [65]) and the atom coordinate and charge files (pqr files) produced by AmberTools19 cpptraj. PyMol (Schrödinger, LLC) was used to draw molecular models and electrostatic potential was mapped on the solvent excluded surface. The interactions between MCoAEP2 and MCoTI-II during the simulations were computed using an in-house script.

The force field parameters, simulation protocol and simulation frames were submitted to Zenodo public data repository with the DOI 10.5281/zenodo.3621201.

**Reporting summary**. Further information on research design is available in the Nature Research Reporting Summary linked to this article.

## Data availability

RNA-seq data have been deposited in the NCBI-SRA database under the accession code PRJNA531039. GenBank accessions for AEPs gene sequences are as follows: *MCoAEP1* (MK770254) and *MCoAEP2* (MK770255). The force field parameters, simulation protocol and simulation frames were submitted to Zenodo public data repository with the DOI 10.5281/zenodo.3621201. All other data supporting the findings of this study are included in the manuscript or supporting information files.

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

## Acknowledgements

This work was supported by Australian Research Council (ARC) grants DP150100443 and DP200101299. D.J.C. is an ARC Australian Laureate Fellow (FL15100146). L.Y.C. was supported by the Advance Queensland Women's Academic Fund (WAF-6884942288). We acknowledge Mr Olivier Cheneval for peptide synthesis and Mr Alun Jones for the assistance with mass spectrometry.

## Author contributions

J.D., L.Y.C., T.D. and D.J.C. designed research. J.D., K.Y., F.R. and F.L. produced the recombinant enzymes. J.D. carried out cyclization assays, kinetic analysis, cyclotide precursors purification and HPLC co-elution studies. L.Y.C. and J.D. carried out NMR. Q.K performed the molecular modeling. E.K.G. contributed to the analysis of AEP transcriptome data. A.G.P. contributed to the mass spectrometry analysis. All authors contributed to writing the paper.

## Competing interests

The authors declare no competing interests.
