## [Peer Review File · Nature Communications]

Reviewers' comments:

Reviewer #1 (Remarks to the Author):

This manuscript by Du et al. reported two new asparaginyl endopeptidases (MCoAEP1 and MCoAEP2) and biochemical characterisation of MCoAEP2. The computational part has been carefully designed and well-executed. The topic would be of high interests for Nat Comm readers. However, I do have a few comments for consideration.

Protonation state: Asp77, Glu216 and Cys218 were kept unprotonated, contrary to what PROPKA predicted. For the catalytic Cys218, the authors rationalised with the problem in predicting Cys pKa as previously shown in Ref. 61. While I agreed with the authors such an error will propagate around the active site, I would think that this error would underestimate the pKas of Asp77 and Glu216 so they would be protonated even more. The authors are strongly recommended to provide an atomistic picture of the active and provide the modelled structure somehow readers/reviewers can access. By looking at Km between MCoTII-II-DAL vs -NAL, it does seem to suggest that the negatively charged active site preferably binds better the neutral peptide than the -ve peptide. Again, readers would be interested to see the interactions between MCoAEP2 and the P1 site.

Electrostatics for preferential processing: This hypothesis is very interesting! I would be interested to see the average and fluctuations of the electrostatic potential along the apo MD trajectories (Fig. 4).

For the reproducibility, the relevant force field parameters and scripts should be provided on a public server.

Reviewer #2 (Remarks to the Author):

The manuscript from Durek and Craik describes the identification and biochemical characterization of a new member of the Asn endopeptidase (AEP) family that is involved in the biosynthesis of a cyclic peptide inhibitor of trypsin. The manuscript is fairly straightforward and the experimental logic is easily followed. While the context is certainly well above the bar for publication in a high profile journal such as Nature Comms., there are some concerns that must be addressed before manuscript may be accepted. I look forward to reviewing a revised version of the current manuscript for Nature Comm. in which these concerns have been addressed.

Major concerns:

1. The natural substrate for the McoAEPs are precursor peptides in which multiple core sequences are embedded. While the authors claim that only minimal residues that are at the N- and C-terminus of the core are enough for cleavage, there is a major concern that additional residues at both termini may be required for efficient catalysis. At the very least, the authors ought to test at least one peptide that contains the entire 20 residue N-terminal and 16 residue C-terminal extension to demonstrate that their statement on Page 5, line 98 extends to this AEP as well.
2. Likewise, kinetics analysis should be carried out for at least one full-length TI-2 unit to rule out any influence of the N- and C-terminal extensions on substrate recognition.
3. A very understated point in the manuscript is that these enzymes seem to be "scaffold specific" in the sense that only peptides that have the correct three dimensional structure are substrates (for example, OaAEP does not process the substrate for McoAEP). I think there is more to Figure 4 than simple difference in charge interaction. The binding cavities look entirely different. To me, it does not look as if the MCoTI-2 substrate could be docked onto the OaAEP cavity without

significant steric clashes. If this is correct, the section of the manuscript needs to be re-written.

Minor points:

1. Please use the more generic terms "leader peptide" and "follower peptide" (rather than N- and C-terminal pro-peptide) to describe the flanking sequences.
2. I could not find a sequence of the grafted cyclic peptide McoSST-01. Please add this to Figure 5.
3. Please label the two lanes in Supplemental Figure 3a (right side).
4. Please add experimentally determined masses for the enzyme cyclized and chemically synthesized species in Figure 5, panel a.

Reviewer #3 (Remarks to the Author):

The work is characterization of new asparaginyl endopeptidase (MCoAEP2) from *Momordica cochinchinensis*. The AEP has peptidase and peptide ligation activities, and involved in the production of cyclic peptide of trypsin inhibitor (MCoTI-II). MCoAEP2 can be classified as a peptidase-type AEP rather than ligase-type, according to the amino acid signatures where the authors proposed in former works. Therefore, authors suggest differences of surface charges of substrate binding pocket in enzyme is involved in the difference of cyclization rate, and difference of P1 amino acids among the NTPP and CTPP will be required for proper cyclization of MCoTI-II. With these data, authors suggest coevolution of substrates and enzymes in each species. The manuscript is well written. Peptide ligation mechanism of AEP is a hot topics and the work provides deep insight into the molecular mechanism of them. Indeed, it is not new that the identification of AEPs with both peptidase and ligase activity, but authors put this as a manuscript title.

I could not follow the molecular bases of MoCoTI-II cyclization in the manuscript; Does MoCoTI-II have a nature of cyclization when cleaved CTPP by any AEP, or MoCoAEP2 has a tendency of ligation activity over peptidase when it works for any substrate? Related to this, what will happen if MoCoTI-II-DAL or MoCoTI-II-NAL is treated with typical peptidase-type AEP?

Line 132. "Compared to other AEPs, MCoAEP1 and MCoAEP2 have a sequence identity...". Could it be possible to add information of expression pattern of MCoAEP1 and MCoAEP2 in plant tissues? Are they co-regulated with MCoTI-II? It will be not important including such information here, but will be helpful for the understanding of MCoTI-II production mechanisms in plant.

Line 192. "we modeled a covalently linked MCoTI-II substrate (which was linked via a thioester bond to the Cys218 side chain of MCoAEP2)". Could you show Cys218 in the Fig. 4? It is difficult to judge the methodologically right without the information.

Line 201. "We also found the molecular surface of MCoAEP2 that interacts with the MCoTI-II substrate is mostly electronegative....." and Fig. 4. Is it possible to explain the changes of pH optimal in peptidase and ligase activity with surface charges?

Line 202. Fig. 4. "in line with the global positive charge of MCoTI-II"
Could the author present data showing positive charge of MCoTI-II in the 3D modeling? Could the author show comparison in the binding sites to AEP between MCoTI-II and kB1?

Line 330. "This finding suggests that AEPs may have co-evolved with their substrates and may offer new strategies for identifying novel AEPs with, for example, altered substrate preferences for

biotechnological applications.”

Is it actually the case? Could authors show evidence of coevolution? I think it is easy for authors to make 3D models of APEs and CCK peptides from other species, and can discuss the possibility of coevolution.

Line 317. “Interestingly, MCoAEP2 is also predicted to be a hydrolase based on sequence similarities in the MLA, LAD1, and LAD2 regions alone Therefore, other factors beside the abovementioned regions must contribute to the preferential activity of AEPs.” MCoAEP2 breaks the rules completely. Again, is it possible to explain how peptide-ligation occurs with MCoAEP2 by comparing structures of AEPs (maybe 3D)?

Line 387. “We note that the NTPP of MCoTI precursors (DIN) is highly conserved...suggesting a selective pressure for a preserved processing mechanism... Additionally, the NTPP processing site of all natural MCoTI precursors is always an Asn; this N-terminal Asn is processed faster by AEPs than Asp, which is always present at the C-terminal cyclization site.”

The description implies that the order of NTPP and CTPP cleavage is important for peptide cyclization, and that makes selective pressure of Asn and Asp at P1 site. If so, peptide cyclization rate will be affected if the authors use a P1-swapped peptide, like “DID-MCoTI-II-NAL”. Is it possible to do such experiment to validate the hypothesis? (However, I am sure that the authors need to block the amino group in N-terminal in this experiment.)

We thank the reviewers for their helpful comments. In the revised manuscript and supplementary information, we have addressed all of the Reviewers' comments, as detailed in our response below.

Reviewer#1

Comment 1. Protonation state: Asp77, Glu216 and Cys218 were kept unprotonated, contrary to what PROPKA predicted. For the catalytic Cys218, the authors rationalised with the problem in predicting Cys pKa as previously shown in Ref. 61. While I agreed with the authors such an error will propagate around the active site, I would think that this error would underestimate the pKas of Asp77 and Glu216 so they would be protonated even more. The authors are strongly recommended to provide an atomistic picture of the active and provide the modelled structure somehow readers/reviewers can access. By looking at Km between MCoTII-II-DAL vs -NAL, it does seem to suggest that the negatively charged active site preferably binds better the neutral peptide than the -ve peptide. Again, readers would be interested to see the interactions between MCoAEP2 and the P1 site.

Response: The well-accepted mechanism of cysteine proteases implies that Cys218 of AEP should be deprotonated, but the theoretical predictions of PROPKA predicted that this Cys218 to be protonated at pH 5.5. As we mentioned in the text and as quoted by the reviewer, PROPKA has been shown to perform poorly in predicting the pKa of active site cysteine residues. We agree with the reviewer that the deprotonation of Cys218 should further push the protonation equilibrium of Asp77 and Glu216 towards their protonated state. We have therefore repeated all our simulations using these protonation states (2 μ s in total), updated Figure 4, the text and added several supplementary figures (Supplementary Figures 7, 8 and 9). Figure 4 e provides a view of the active site of MCoAEP2 in the apo state. Figure 4 f shows a view of the active site in an intermediate state in which the side chain of Cys218 is covalently linked via a thioester bond to MCoAEP2.

Comment 2. Electrostatics for preferential processing: This hypothesis is very interesting! I would be interested to see the average and fluctuations of the electrostatic potential along the apo MD trajectories (Fig. 4).

Response: The electrostatic potential computed with APBS uses boundary conditions set at the molecular surface, which is spatially different at each frame of the simulation as the side chains move. It is therefore not possible to define an average electrostatic potential along an MD trajectory. Instead, we have shown the consistency of the electrostatic surface during 10 MD trajectories of 100 ns (1 μ s in total) by clustering the frames extracted over all simulations and drawing the electrostatic potential on the surface of the centroid frame of each cluster. As shown in the new Supplementary Figure 8, the electrostatic potential of these centroid frames is similar and the MCoTI-II binding sites in these frames are predominantly negatively charged.

Comment 3. For the reproducibility, the relevant force field parameters and scripts should be provided on a public server.

Response: We have submitted the topology and simulation protocol, as well as starting frames, final frames, centroid frames to Zenodo (<https://zenodo.org/>) with the DOI [10.5281/zenodo.3621201](https://doi.org/10.5281/zenodo.3621201). This information has been added to the manuscript at the end of the Methods section.

Reviewer#2

Comment 1. The natural substrate for the MCoAEPs are precursor peptides in which multiple core sequences are embedded. While the authors claim that only minimal residues that are at the N- and C-terminus of the core are enough for cleavage, there is a major concern that additional residues at both termini may be required for efficient catalysis. At the very least, the authors ought to test at least one peptide that contains the entire 20 residue N-terminal and 16 residue C-terminal extension to demonstrate that their statement on Page 5, line 98 extends to this AEP as well.

Response: We thank the reviewer for this suggestion. We have recombinantly produced in *E. coli* the requested ‘native’ precursor peptide corresponding to unit 3 of TIPTOP2 from *M. cochinchinensis*. The 66 amino acid protein was expressed as a fusion protein and after removal of tags contained 16 N-terminal and 16 C-terminal propeptide residues. Correct folding of the TI domain was confirmed by NMR (we have updated Supplementary Figure 15 accordingly).

Comment 2. Likewise, kinetics analysis should be carried out for at least one full-length TI-2 unit to rule out any influence of the N- and C-terminal extensions on substrate recognition.

Response: We have compared the 66 AA TIPTOP2 unit 3 with the shortened 39 AA DIN-MCoTI-II-DAL precursor in enzyme cyclization assays using MCoAEP2 (see new Supplementary Figure 13). While a detailed MM kinetic analysis was not possible due to the complexity of the multistep enzymatic reaction and the large number of intermediates (as we have previously noted for DIN-MCoTI-II-DAL), we were able to quantify the precursor peptides via MRM-MS and monitor substrate depletion over time under identical enzyme assay conditions. The kinetic data show that TIPTOP2 unit 3 is processed marginally slower than DIN-MCoTI-II-DAL with observed half-lives of 29 min and 10 min, respectively. In kinetics these rate differences are negligible and thus, the additional residues in the native precursor peptides do not significantly affect the cyclization kinetics of MCoAEP2. Similar findings have been observed for other AEPs (OaAEP1_b) and kalata B1 precursors (see Conlan et al., *J Biol Chem* 287, 28037-28046 (2012)). We have added a note to the main manuscript summarizing these findings (page 13).

We analyzed the processing of a recombinantly produced TIPTOP2 MCoTI-II precursor carrying the full-length leader and follower sequences by MCoAEP2 and compared it to DIN-MCoTI-II-DAL under identical conditions. Both precursors were processed with similar kinetics, indicating

that the additional leader and follower residues in the native precursors do not significantly affect the cyclization kinetics (Supplementary Fig. 13).”

Comment 3. A very understated point in the manuscript is that these enzymes seem to be "scaffold specific" in the sense that only peptides that have the correct three dimensional structure are substrates (for example, OaAEP does not process the substrate for McoAEP). I think there is more to Figure 4 than simple difference in charge interaction. The binding cavities look entirely different. To me, it does not look as if the MCoTI-2 substrate could be docked onto the OaAEP cavity without significant steric clashes. If this is correct, the section of the manuscript needs to be re-written.

Response: The binding mode of MCoTI-II on MCoAEP2 was not stable during the 1 μ s simulation. To illustrate that there is not a single binding orientation that could capture the binding interaction, we have added a Supplementary Figure 7, which shows various binding orientations observed during the 1 μ s combined simulation time. The C-terminus of MCoTI-II, despite being covalently tethered to Cys218 of MCoAEP2, is indeed flexible and relatively long, enabling considerable freedom of orientation MCoTI-II. The binding surface of OaAEP1_b and MCoAEP2 are similarly relatively flat, and MCoTI-II could be similarly tethered to the active site Cys of OaAEP1_b without creating steric clashes.

Comment 4. Please use the more generic terms "leader peptide" and "follower peptide" (rather than N- and C-terminal pro-peptide) to describe the flanking sequences.

Response: We have changed the terms to “leader peptide” and “follower peptide” in both manuscript and supplementary files as suggested by the reviewer (highlighted in yellow).

Comment 5. I could not find a sequence of the grafted cyclic peptide McoSST-01. Please add this to Figure 5.

Response: The sequence of grafted cyclic peptide MCoSST-01 is shown in Fig. 5b (bottom panel, X-axis label).

Comment 6. Please label the two lanes in Supplemental Figure 3a (right side).

Response: We have labelled the two lanes and added explanations in the figure caption.

Comment 7. Please add experimentally determined masses for the enzyme cyclized and chemically synthesized species in Figure 5, panel a.

Response: We have added mass in the panel a of Fig. 5.

Reviewer#3

The manuscript is well written. Peptide ligation mechanism of AEP is a hot topics and the work provides deep insight into the molecular mechanism of them. Indeed, it is not new that the identification of AEPs with both peptidase and ligase activity, but authors put this as a manuscript title.

Comment 1. I could not follow the molecular bases of MCoTi-II cyclization in the manuscript; Does MCoTi-II have a nature of cyclization when cleaved CTPP by any AEP, or MCoAEP2 has a tendency of ligation activity over peptidase when it works for any substrate? Related to this, what will happen if MCoTi-II-DAL or MCoTi-II-NAL is treated with typical peptidase-type AEP?

Response: When MCoTI-II-DAL was incubated with OaAEP1b (a different ligase AEP from another plant species) the cyclization rate of MCoTI-II turned out to be extremely slow which prompted us to identify the native AEP catalyzing MCoTI processing (this study).

We recombinantly expressed three different AEPs that have been previously shown to be predominantly peptidase AEPs (HaAEP1 from *Helianthus annuus*, AtAEP2 from *Arabidopsis thaliana* and human legumain). When tested against MCoTI-II-DAL and MCoTI-II-NAL these enzymes were generally poor at recognizing MCoTI-II-DAL (the native substrate), but they showed cyclization ability in terms of MCoTI-II-NAL substrate. This unexpected cyclization capability in these ‘peptidase-type’ AEPs further confirms our hypothesis, that the outcome of the enzyme-catalyzed reaction is, at least in part, substrate-dependent.

We have added a new Supplementary Figure 11 and added the following section to the manuscript:

Peptidase-type AEPs are weak at recognizing MCoTI-II-DAL

We also tested the synthetic MCoTI precursors against the well characterized AEPs from *Helianthus annuus* (HaAEP1), *Arabidopsis thaliana* (AtAEP2) and human legumain, which have all been characterized as predominant peptidase type AEPs. While the native-like MCoTI-II-DAL is a very poor substrate for these enzymes, MCoTI-II-NAL was more efficiently processed yielding correctly cyclized product (Supplementary Fig. 11). This unexpected cyclization capability in these ‘peptidase-type’ AEPs suggest that the outcome of the enzyme-catalyzed reaction is at least in part substrate-dependent.

Comment 2. Line 132. “Compared to other AEPs, MCoAEP1 and MCoAEP2 have a sequence identity...”. Could it be possible to add information of expression pattern of MCoAEP1 and MCoAEP2 in plant tissues? Are they co-regulated with MCoTI-II? It will be not important including such information here, but will be helpful for the understanding of MCoTI-II production mechanisms in plant.

Response: We agree with the reviewer that tracing the co-expression of MCoAEPs and their substrates would be interesting as a way to add biological context. In brief, we cannot see any specific pattern of expression for either MCoAEP from our read coverage after mapping reads to the assembled transcriptome. There is some expression in both across all organs sampled whereas the TIPTOP2 substrate seems restricted to seed based on read mapping. We would like to note that our RNA-seq data was obtained from a single replicate each from seed, shoot (leaf and apical meristem), flower, and root. This was done to maximise collection of sequences containing the entirety of the plants transcriptomic diversity, not as a quantitative measurement. Therefore, we are wishing to remain conservative regarding claims about expression patterns.

Comment 3. Line 192. “we modeled a covalently linked MCoTI-II substrate (which was linked via a thioester bond to the Cys218 side chain of MCoAEP2)”.

Could you show Cys218 in the Fig. 4? It is difficult to judge the methodologically right without the information.

Response: We added a sub-panel f in Figure 4 showing the covalent link between MCoTI-II C-terminus and MCoAEP2 Cys218 side chain. The position of the Asx binding pocket is highlighted on panels a to c of Figure 4 by a yellow circle. Additionally, the molecular models can now be downloaded from Zenodo (doi: [10.5281/zenodo.3621201](https://doi.org/10.5281/zenodo.3621201)).

Comment 4. Line 201. “We also found the molecular surface of MCoAEP2 that interacts with the MCoTI-II substrate is mostly electronegative.....” and Fig. 4.

Is it possible to explain the changes of pH optimal in peptidase and ligase activity with surface charges?

Response: The pH dependence of the electrostatic surface of MCoAEP2 is now presented in the new Supplementary Figure 9 for pH 3.5 to 7.5. In these computations we assumed that Cys218 would remain deprotonated as it is a requirement for activity. The potential mapped on the surface of the binding site of MCoTI-II does not seem to vary extensively in this pH range, suggesting that the electrostatic potential of binding site is not determinant of the optimal pH for processing MCoTI-II. We have added a comment on page 10-11.

Comment 5. Line 202. Fig. 4. “in line with the global positive charge of MCoTI-II”

Could the author present data showing positive charge of MCoTI-II in the 3D modeling? Could the author show comparison in the binding sites to AEP between MCoTI-II and kB1?

Response: We have added a sub-panel d for Figure 4 showing the electrostatic potential of the surface of MCoTI-II that is suggested to be at the interface with MCoAEP2. The electrostatic potential mapped on this surface of MCoTI-II is electropositive, in agreement with our theory of electrostatic potential driving the selectivity for MCoAEP2.

Because of the covalent link between the C-terminus of the substrate peptide and Cys218, the interaction site of kB1 and MCoTI-II should be highly similar. Figure 4b and 4c already displays a comparison of these binding sites between MCoAEP2 (binding site of MCoTI-II) and

OaAEP1_b (binding site of kB1).

Comment 6. Line 330. “This finding suggests that AEPs may have co-evolved with their substrates and may offer new strategies for identifying novel AEPs with, for example, altered substrate preferences for biotechnological applications.”

Is it actually the case? Could authors show evidence of coevolution? I think it is easy for authors to make 3D models of AEPs and CCK peptides from other species, and can discuss the possibility of coevolution.

Response: Upon review of our manuscript's message, we feel that it is not in the scope of this work to test evolutionary claims. The passage has been replaced to take on a more conservative statement about evolutionary claims:

With this finding we hypothesize that AEPs may have co-evolved with their substrates. Testing of this hypothesis in future work may provide data that could underpin strategies for identifying novel AEPs with, for example, altered substrate preferences for biotechnological applications. The activities of these enzymes against cyclotide precursors from other families remain to be determined.

Comment 7. Line 317. “Interestingly, MCoAEP2 is also predicted to be a hydrolase based on sequence similarities in the MLA, LAD1, and LAD2 regions alone Therefore, other factors beside the abovementioned regions must contribute to the preferential activity of AEPs.” MCoAEP2 breaks the rules completely. Again, is it possible to explain how peptide-ligation occurs with MCoAEP2 by comparing structures of AEPs (maybe 3D)?

Response: The mechanism of peptide-ligation taking place is expected to be similar to other reported AEPs (cyclase/ligase). The predicted regions (MLA, LAD1, LAD2 etc.) conclude most cases of these ligases, however our finding did not follow these rules, suggesting that the concluded rules did not apply to all situations. The precise role these regions play on the atomic level is not very well understood, which is further compounded by the lack of experimentally determined AEP-substrate complex structures (all AEP structures determined so far are in the apo enzyme state or of AEPs where the active site has been blocked with an inhibitor). Our modelling study provides the first insights into how substrate recognition may be governed by MLA, LAD1 and LAD2 but in our opinion experimentally determined structures of higher resolution are needed to confidently make mechanistic claims.

Comment 8. Line 387. “We note that the NTPP of MCoTI precursors (DIN) is highly conserved...suggesting a selective pressure for a preserved processing mechanism... Additionally, the NTPP processing site of all natural MCoTI precursors is always an Asn; this N-terminal Asn is processed faster by AEPs than Asp, which is always present at the C-terminal cyclization site.”

The description implies that the order of NTPP and CTPP cleavage is important for peptide cyclization, and that makes selective pressure of Asn and Asp at P1 site. If so, peptide cyclization rate will be affected if the authors use a P1-swapped peptide, like “DID-MCoTI-II-NAL”. Is it possible to do such experiment to validate the hypothesis? (However, I am sure that the authors need to block the amino group in N-terminal in this experiment.)

Response: Yes, we agree with the referee that the order of NTPP and CTPP processing is extremely important and thank them for this valuable suggestion. We have prepared by chemical synthesis two P1-swapped MCoTI precursor substrates, i.e. DID-MCoTI-II-NAL and AcDID-MCoTI-II-NAL as suggested by the referee and have compared their processing by MCoAEP2 to DIN-MCoTI-II-DAL (the native substrate). The results have been added to Supplementary Figure 12.

AcDID-MCoTI-II-NAL was predominantly cleaved/hydrolyzed at the C-terminal processing site by MCoAEP2 resulting in linear AcDID-MCoTI-II-N (Supplementary Fig. 12c). This finding is in agreement with our observation that P1 Asn is processed significantly faster than Asp. The preferential processing at the C-terminal site and the blocked N-terminus prevented any further enzyme processing/cyclization under these conditions. Similarly, DID-MCoTI-II-NAL was preferentially processed at the C-terminal processing site (hydrolysis of Asn-Ile peptide bond). However a small fraction was able to undergo transpeptidation to give a mis-cyclized product, cyc[DID-MCoTI-N] (Supplementary Fig. 12d). These results further confirm our hypothesis that the order of processing is important for native MCoTI-II cyclization in planta and that the placement/conservation of Asn/Asp at the N- and C-terminal processing sites is not random, but likely governs the processing at these sites.

We have amended the main manuscript including results section at pages 13,14 and 20 and methods section at page 25.

REVIEWERS' COMMENTS:

Reviewer #1 (Remarks to the Author):

The authors have addressed my concerns, therefore I recommend acceptance.

Reviewer #2 (Remarks to the Author):

The authors have addressed each of the points that were raised during the initial review of this manuscript. This reviewer believes that the changes have made for a manuscript that is much more accessible to the broad readership of Nature Communications. I heartily endorse publication of this revised manuscript.

Reviewer #3 (Remarks to the Author):

All responses from the authors are clear, with no question in the paper.

We thank the reviewers for their helpful comments that have improved our manuscript.

REVIEWERS' COMMENTS:

Reviewer #1 (Remarks to the Author):

The authors have addressed my concerns, therefore I recommend acceptance.

Reviewer #2 (Remarks to the Author):

The authors have addressed each of the points that were raised during the initial review of this manuscript. This reviewer believes that the changes have made for a manuscript that is much more accessible to the broad readership of Nature Communications. I heartily endorse publication of this revised manuscript.

Reviewer #3 (Remarks to the Author):

All responses from the authors are clear, with no question in the paper.